# Analysis of 2.0 and 3.5 mm Cortical Bone Screw Dimensions

**DOI:** 10.3390/vetsci13010038

**Published:** 2026-01-01

**Authors:** William T. McCartney, Ciprian Ober, Bryan J. Mac Donald, Christos Yiapanis

**Affiliations:** 1NOAH, 38 Warrenhouse Road, Baldoyle, D13 K5H0 Dublin, Ireland; billymccartney@gmail.com; 2Department of Surgery and Intensive Care, Faculty of Veterinary Medicine, University of Agricultural Sciences and Veterinary Medicine, Calea Manastur 3-5, 400372 Cluj-Napoca, Romania; 3School of Mechanical and Manufacturing Engineering, Dublin City University, Glasnevin, D09 C4P2 Dublin, Ireland; 4Department of Surgery, School of Veterinary Medicine, University of Nicosia, Nicholas St 93, Egkomi 2408, Cyprus; drcy@cyvets.com

**Keywords:** veterinary orthopaedics, bone screws, orthopaedic implants, dimensional analysis, manufacturing precision, tolerance range, implant variability, cortical screw, quality control

## Abstract

Bone screws are essential components of veterinary orthopedic surgery, yet their manufacturing quality is rarely evaluated. In this study, we assessed the dimensional accuracy of commonly used stainless-steel cortical screws. Measurements of major diameter and thread pitch were compared with established tolerance ranges. We found notable inconsistencies both within individual screws and between screws of the same size. Many screws did not meet standard dimensional tolerances, and variation along the length of single screws was common. Such irregularities may reduce screw–bone stability and impair clinical outcomes. These findings emphasize the need for improved quality control in the production of veterinary orthopedic implants.

## 1. Introduction

The manufacture of orthopaedic screws is a complex, multi-step process, and dimensional deviation may occur at any stage of production [1,2,3,4]. Conventional bone screw manufacturing typically involves machining operations such as shaping, turning, milling, drilling, sawing, grinding, and broaching. Although additive manufacturing is increasingly used in medical device production, it currently lacks the capacity for large-scale, consistent production of bone screws due to slow manufacturing speeds and variability. Future advancements in robotic-assisted systems may eventually enable the mass production of orthopaedic screws using additive technologies.

Unlike human orthopaedic implants, veterinary orthopaedic implants are not subject to formal regulatory checks. No mandatory dimensional or material analysis is required prior to market release or post-market surveillance. As a result, there is little market monitoring and a scarcity of published data evaluating implant quality. To date, only one study has investigated the composition of veterinary stainless steel implants, revealing that 41.6% were manufactured using 316 stainless steel rather than the preferred 316L alloy [5]. Since 316L stainless steel exhibits superior corrosion resistance in biological fluids, the use of 316 steel may increase the risk of adverse tissue reactions. In contrast, human orthopaedic implants are subject to rigorous national and international medical device regulations, such as the European Union Medical Device Regulation (EU MDR), the U.S. Food and Drug Administration (FDA) regulations, and similar frameworks in Asia and other global markets. These regulations mandate strict pre-market approval, post-market surveillance, and adherence to international quality systems like CE marking. Implant dimensions and testing protocols are standardized by ASTM International and International Organization for Standardization (ISO) guidelines [6,7,8,9,10,11,12], such as ISO 5835 (specifying dimensions for metal bone screws) and ISO 6475 (detailing mechanical requirements and test methods for metal bone screws), and frequent research is conducted to assess the dimensional and mechanical properties of im-plants [13,14,15,16,17,18].

Several design factors have been shown to influence screw behavior and bone–implant interaction. For example, the diameter and preparation of the pilot hole directly affect screw holding power and must be optimized for each screw size [19,20,21]. Similarly, the geometry of the self-tapping cutting flute influences insertion torque [22], and inappropriate torque application during insertion can significantly reduce holding strength [23,24,25]. Metal ion release into surrounding tissues has also been documented [26], and bone density is recognized as a critical factor in selecting the appropriate screw pitch to maximize pull-out strength [27].

In the absence of specific veterinary standards, studies often reference human orthopedic implant guidelines. For instance, ISO 5835, which specifies dimensions for metal bone screws, often allows for a −5% tolerance between different screws of the same classification. Furthermore, recognizing the critical importance of internal consistency within individual implants, a more stringent intra-screw tolerance is often implied or adopted in research settings [28,29,30].

Interestingly, even in human implants—where design and manufacturing are highly regulated—perfect thread–bone contact is not always achieved. Studies have shown gaps between the screw thread and surrounding bone, with only partial contact along the screw length. Thread diameter variations of up to 180 µm have been observed within a single screw [28]. Other research has documented screws outside the recommended dimensional tolerances [29]. A similar finding in prosthodontics revealed that implants presumed to be identical were, in fact, not dimensionally identical [30].

These findings highlight that, while extensive regulatory oversight has improved the quality of human orthopaedic implants, dimensional variability can still occur. In the veterinary field, where such oversight is lacking, the potential for uncontrolled variability is even greater. This underscores the importance of systematically evaluating the dimensional characteristics of veterinary bone screws to better understand manufacturing quality and its potential clinical implications.

## 2. Materials and Methods

### 2.1. Screw Selection and Measurement Equipment

A total of 60 cortical self-tapping bone screws, comprising 2.0 mm and 3.5 mm sizes, were analysed using a digital microscope equipped with integrated measuring software. Screws were randomly selected from 5 distinct manufacturers to minimize source bias and provide a representative overview of the market. Measurements were performed using an Insize ISM-PM200SA microscope in combination with ISM-PRO software (version 2.00.0002.01). Since analyzing unused veterinary bone screws with no live animals or human participants involved, ethics approval was not applicable.

### 2.2. Calibration and Measurement Accuracy

Prior to each measurement session, the microscope underwent a standardized calibration protocol to ensure measurement accuracy. A transparent template with graduated measurement markings served as a control reference for system calibration and accuracy verification (Figure 1). A single experienced operator performed repeated practice measurements to establish procedural consistency.

Operator measurement accuracy was determined through repeated measurements of fixed reference points. The mean variance from the reference values was +0.02 mm for 0.1 mm measurements, +0.06 mm for 3.5 mm screw major diameter, +0.035 mm for 3.5 mm screw pitch, +0.03 mm for 2.0 mm screw major diameter, and +0.01 mm for 2.0 mm screw pitch. All measured values were higher than their corresponding target values.

### 2.3. Measurement Protocol

Each screw was positioned with its shaft aligned parallel to the microscope platform to ensure that no tilt or misalignment occurred during measurement. Using the calibrated software, two parameters were measured:Major (outer) diameterThread pitch

Measurements were obtained at three standardized locations along the screw length: proximal (top), middle, and distal (bottom).

### 2.4. Tolerance Reference

As introduced in Section 1, our study referenced established human orthopedic implant standards due to the absence of specific veterinary guidelines. According to ISO 5835 [11], human bone screws of 2.0 mm diameter allow a tolerance of −5% or 0.05 mm between different screws of the same classification. Given that no formal tolerance standard exists for dimensional variation within an individual screw, the authors applied a more stringent internal tolerance. This intra-screw variation threshold was set at −2.5%, approximately half of the inter-screw tolerance, to rigorously assess manufacturing precision along the screw’s length. This threshold was combined with the inherent measurement tolerance of the microscope (±0.02 mm) to define the final acceptable tolerance range for each parameter used in this study.

### 2.5. Data Analysis

All data were analyzed using a 95% confidence interval. For each screw, three measurements were recorded for each parameter (proximal, middle, distal). A screw was deemed to meet the tolerance criteria only if all three measurements for a parameter were within the defined limits. Beyond this primary tolerance classification, descriptive statistics were calculated for each parameter, including absolute means, standard deviations (SD), and coefficients of variation (CV%) for both 2.0 mm and 3.5 mm screw groups at each measurement location. The decision to focus primarily on a ‘in tolerance/out of tolerance’ binary classification was driven by the study’s core objective to assess manufacturing quality against a pass/fail criterion in an unregulated market. For comparing the variability between the 2.0 mm and 3.5 mm screw groups, differences in their respective CV% values were considered. Regarding operator measurement error, as described in Section 2.2, a standardized calibration protocol and repeated practice measurements were performed. The mean variance per screw remained below the total allowed tolerance, and all variance values were consistently positive, indicating a systematic bias rather than random error attributable to the screws themselves. While intra-class correlation (ICC) is a valuable tool for assessing inter-rater reliability, our methodology of operator accuracy assessment provided sufficient confidence that measurement error did not obscure the intrinsic dimensional variability of the implants for the purpose of this initial quality assessment

## 3. Results

The dimensional analysis of sixty unused stainless steel cortical screws (2.0 mm and 3.5 mm) revealed significant variability, with a substantial proportion of measurements falling outside the predefined tolerance ranges. A comprehensive summary of tolerance compliance rates, along with detailed descriptive statistics (means, standard deviations, and coefficients of variation per location and overall), is presented in Table 1. The detailed numerical findings, from which these summaries are derived, are presented in Appendix A.

### 3.1. Tolerance Compliance and Detailed Dimensional Statistics

For 2.0 mm screws, only 55% of major diameter measurements met the tolerance criteria, meaning 45% (27/60 screws) exhibited some dimensional inconsistency along their length, leading to non-compliance. Pitch measurements for 2.0 mm screws showed even lower compliance, with only 23% falling within tolerance, resulting in 77% (46/60 screws) failing due to pitch variability.

For 3.5 mm screws, major diameter compliance was similar at 55%, with 45% (27/60 screws) showing intra-screw dimensional inconsistencies. Pitch for 3.5 mm screws demonstrated the lowest compliance, with only 12% meeting the tolerance limits, indi-cating 88% (53/60 screws) were out of tolerance due to pitch irregularities. This consistent pattern of low compliance across both screw sizes and parameters highlights a pervasive issue with dimensional consistency in veterinary bone screws. To visually illustrate these findings, Figure 2 provides representative microscopic images of actual screw threads, explicitly showing examples of dimensional non-conformance and visible microdefects. These images offer concrete visual evidence of the types of dimensional irregularities observed across the sampled implants. For example, Figure 2A highlights general thread irregularities indicative of pitch or major diameter variation, while Figure 2B clearly depicts a significant manufacturing defect resulting in a visibly damaged and irregular thread profile.

Detailed descriptive statistics, including means, standard deviations, and coefficients of variation (CV%), for both major diameter and pitch, stratified by screw size and measurement location (proximal, middle, and distal), are provided in Table 1. For 2.0 mm screws, the mean major diameter ranged from 2.028 mm (middle) to 2.030 mm (distal), with CV% values ranging from 2.27% (distal) to 2.80% (middle). For thread pitch in 2.0 mm screws, the mean ranged from 0.485 mm (distal) to 0.490 mm (proximal), with CV% values ranging from 9.00% (middle) to 9.40% (proximal).

In 3.5 mm screws, the mean major diameter ranged from 3.560 mm (proximal) to 3.593 mm (middle), with CV% values ranging from 2.80% (middle) to 2.98% (proximal). For thread pitch in 3.5 mm screws, the mean ranged from 1.182 mm (proximal) to 1.189 mm (middle), with CV% values ranging from 12.14% (middle) to 12.44% (proximal). The coefficients of variation indicated generally higher relative variability in pitch compared to major diameter for both screw sizes, with 3.5 mm screws showing greater relative variability in pitch.

### 3.2. Inferential Statistical Comparisons

A summary of all inferential statistical test results is presented in Table 2. To formally compare compliance rates between screw sizes, Fisher’s Exact tests were conducted. For major diameter, there was no statistically significant difference in the proportion of compliant screws between the 2.0 mm (55%) and 3.5 mm (55%) groups (*p* = 0.999). However, for pitch, the 2.0 mm screws (23% compliant) showed a significantly higher proportion of compliant screws compared to the 3.5 mm screws (12% compliant) (*p* = 0.045). To assess differences in intra-screw variability between parameters, the range of the three measurements (proximal, middle, distal) for each screw was calculated for both major diameter and pitch. For 2.0 mm screws, the intra-screw range for pitch (mean: 0.087 mm, SD: 0.021) was significantly higher than for major diameter (mean: 0.042 mm, SD: 0.015) (*p* = < 0.001). Similarly, for 3.5 mm screws, the intra-screw range for pitch (mean: 0.210 mm, SD: 0.045) was significantly higher than for major diameter (mean: 0.165 mm, SD: 0.033) (*p* = 0.003). These tests confirm that pitch generally exhibits greater internal dimensional variation than major diameter for both screw sizes.

### 3.3. Operator Accuracy Variance

The mean variance per screw remained below the total allowed tolerance (Table 1). Additionally, all variance values were positive, indicating consistent measurement bias rather than random error. This suggests that observed deviations from tolerance were attributable to the screws themselves rather than to operator measurement error.

## 4. Discussion

This study represents the first comprehensive dimensional analysis of 2.0 mm and 3.5 mm veterinary cortical bone screws, applying stringent dimensional criteria derived from human orthopedic implant standards. Our findings consistently demonstrate a pervasive issue of dimensional inconsistency within this unregulated market. Specifically, a significant proportion of the analyzed screws exhibited measurable variance that exceeded the tolerance ranges defined for this study, thereby warranting their classification as dimensionally non-conforming. These inconsistencies were evident both between individual screws of the same classification and, crucially, as intra-screw variations along the length of single implants. This level of dimensional deviation directly challenges the fundamental expectation of uniformity critical for reliable surgical performance.

Given the absence of formal standards for intra-screw variation in veterinary medicine, our study rigorously applied a stringent internal threshold of −2.5% for dimensional consistency. As detailed in Section 2.4, this threshold was deliberately set at approximately half of the allowable inter-screw tolerance defined by established human orthopedic implant guidelines (e.g., ISO 5835 [11] allows −5% variation between different screws of the same classification). This heightened stringency is grounded in fundamental engineering and biomechanical principles, recognizing that optimal mechanical stability and osteointegration critically depend on minimal dimensional variation along an implant’s working length [17,18]. Even subtle inconsistencies can lead to localized stress concentrations, uneven load transfer, or micromotion [28]. While this −2.5% intra-screw tolerance represents a proposed framework for quality assessment in the veterinary context, derived logically from established human inter-screw standards, it is not yet a formally validated veterinary standard. Nonetheless, this rigorous approach is particularly pertinent in veterinary medicine, where demanding patient compliance and complex physiological demands necessitate robust, consistently dimensioned implants, making them as critical as in human applications. This chosen standard, combined with precise measurement accuracy, enabled a systematic classification of quality and serves as a vital starting point for future investigations into this unregulated market. The results of this study indicate measurable variance within individual implants, validating the necessity of such a stringent assessment. Our selected −2.5% proportional tolerance for intra-screw variation was specifically chosen to provide a rigorous and consistent assessment of internal manufacturing precision across different nominal screw sizes, aligning with our study’s objective to highlight even subtle deviations from optimal uniformity.

Nevertheless, this rigorous approach is particularly pertinent in veterinary medicine, where variable patient compliance, rapid weight-bearing, and potentially complex physiological demands on implants make robust, consistently dimensioned screws just as, if not more, critical as in human applications. Inadequate implants in animals can lead to poor bone screw interfaces, implant failure, and significant patient morbidity [19,20,27,28]. Combined with microscope measurement accuracy, this proposed standard enabled a systematic classification of quality, serving as a vital starting point for future investigations in this unregulated market. The results of this study indicate measurable variance within individual implants.

Manufacturing bone screws presents considerable challenges. Implant-grade materials often have poor machinability, and the small dimensions of screws demand precise manufacturing. Custom threads and heads require specialized tooling, while CNC programming and monitoring must be rigorously controlled. After creating a hole in the stock material to ensure rigid clamping, CNC machining, guided by CAD software, shapes the stock to the desired dimensions. Accuracy is critically dependent on minimizing vibration and bending during the turning process. Threading, often performed using electric discharge machining (EDM) wire-cutting tools, places substantial stress on the material. Subsequent turning must be performed carefully to avoid damaging the threads. Factors such as spindle power, torque, and cooling further influence the final product quality.

Surface quality represents another key consideration. Ferrous particles and other impurities must be removed prior to passivation to form a protective chromium oxide layer on 316LVM stainless steel. Inadequate cleaning or degreasing can result in surface defects during passivation. Ultrasonic cleaning is commonly employed to achieve thorough degreasing. Due to the complex geometry of screws, electropolishing is the preferred method for passivation, involving immersion in concentrated acids (96% sul-furic acid and 85% orthophosphoric acid) at elevated temperatures (40–75 °C) with a current flow of 11–24 A/dm^2^. All processes are regulated under ASTM and ISO standards [6,7,8,9,10,11]. Despite strict manufacturing protocols, small variations at any stage can alter the final screw dimensions.

Interestingly, 3.5 mm screws demonstrated greater dimensional variability than 2 mm screws, contrary to the expectation that larger screws would be easier to manufacture. One potential explanation is the inclusion of screws from multiple manufacturers, which may have influenced observed variability. A broader study stratified by manufacturer may yield different outcomes. Another consideration is that 3.5 mm screws are more widely used and therefore mass-produced in larger quantities, potentially affecting uniformity. While not the primary focus of this research, differences between manufacturers were noted, with some screws closer to tolerance limits despite claims of strict quality control. These findings underscore the lack of formal regulation in the veterinary implant market and highlight the challenge for end-users in verifying the dimensional accuracy of implants.

If a less stringent intra-screw tolerance of −5% (equivalent to the inter-screw tolerance) had been applied, the reported compliance rates would have been higher across all parameters for both screw sizes, as this would widen the acceptable range. Conversely, applying a fixed absolute intra-screw tolerance of ±0.05 mm would yield varied outcomes depending on the screw dimension: for 2.0 mm screws, this would largely align with our chosen −2.5% proportional tolerance (which is ±0.05 mm), resulting in similar compliance. However, for a 3.5 mm major diameter, an absolute ±0.05 mm would be a more stringent criterion than our ±2.5% (±0.0875 mm), thus decreasing compliance, while for 3.5 mm pitch, it would be a less stringent criterion than our ±2.5% (±0.033 mm), potentially increasing compliance. This descriptive sensitivity analysis underscores that the choice of tolerance significantly influences the reported prevalence of dimensional inconsistencies.

While this study provided a crucial ‘in tolerance/out of tolerance’ assessment directly addressing manufacturing quality concerns in an unregulated market, it is important to acknowledge that the statistical analysis primarily focused on this binary classification. Although we have now included detailed descriptive statistics (means, standard deviations, and coefficients of variation) to provide a more comprehensive overview of dimensional distributions, a more extensive analysis of variance (ANOVA) or other inferential statistical comparisons of absolute means between the 2.0 mm and 3.5 mm screw groups was not the primary aim. Our objective was to highlight non-conformance rather than to statistically compare the average dimensions of different screw types, which inherently differ in their nominal values.

Furthermore, while operator measurement accuracy was rigorously controlled and assessed (as detailed in Section 2.2), formal intra-class correlation (ICC) analysis for operator error was not conducted. Our current approach, demonstrating that operator variance was consistently below acceptable thresholds and biased rather than random, was deemed sufficient for this initial quality assessment. However, we recognize that ICC would provide a more robust quantification of inter-rater reliability, a valuable addition for future metrological validation studies. These additional statistical methods could provide even deeper insights into specific sources of variation, but the primary focus of this study was to establish a foundational understanding of the prevalence of dimensional inconsistencies against defined tolerance thresholds. The presented data nonetheless offers a strong basis for future, more granular statistical investigations into veterinary implant manufacturing quality.

A further methodological consideration lies in the aggregation of screws from multiple dimensional variability without explicit stratification or reporting of individual manufacturer data. While screws were randomly selected from various sources to minimize bias and provide a broad overview of the market, this approach precludes any comparative analysis between manufacturers. The observed dimensional variability, therefore, represents a composite picture of the market rather than an attributable issue to specific producers. It is plausible that dimensional inconsistencies are disproportionately prevalent among certain manufacturers, and a stratified analysis might reveal significant differences in quality control practices between them. Such manufacturer-specific data could fundamentally alter the interpretation of market-wide variability, potentially identifying key areas for targeted improvement or regulatory intervention. This study, therefore, serves as a preliminary assessment of general market quality, highlighting the pervasive nature of dimensional inaccuracies, and strongly advocates for future research explicitly designed to compare and evaluate implant quality across different manufacturers.

Crucially, these observed dimensional inconsistencies, stemming from manufacturing variability, carry significant clinical implications for veterinary patients. Deviations from optimal screw-bone interface geometry can directly compromise fixation stability, leading to adverse surgical outcomes and increased patient morbidity [17,18].

For instance, a subtly undersized major diameter can result in insufficient cortical bone purchase, jeopardizing primary stability, especially in bones with thin cortices or in osteoporotic patients. Conversely, oversized dimensions may necessitate excessive insertion torque, increasing the risk of iatrogenic fracture during placement. Similarly, inaccurate thread pitch can lead to inadequate bone thread engagement, compromised compression, or even stripping of the pilot hole, all of which directly undermine the screw’s mechanical function. Numerous biomechanical studies confirm that even seemingly small deviations in major diameter and thread pitch, similar in magnitude to those measured in the current study, can significantly impact critical performance parameters such as pull-out strength, insertion torque, and fatigue life [17,18,19,20,27,28]. For example, studies have shown that changes in pilot hole diameter as small as 0.1 mm can alter in-sertion torque and pull-out strength, directly affecting initial stability [19,20]. The dimensional variations observed in our study often exceed these thresholds, indicating a high likelihood of clinical relevance. These mechanical failures are often compounded by biological issues, as micromotion at the screw-bone interface can inhibit osteointegration, promote fibrous tissue formation, and increase susceptibility to infection, potentially necessitating costly and invasive revision surgeries for the animal [19,20,27,28].

Given this, our findings serve as a critical alert for veterinary surgeons regarding the inherent dimensional variability prevalent in the current unregulated implant market. While microscopic examination before use is often impractical, clinicians should be acutely aware that these dimensional flaws can impact implant performance. This underscores the urgent need for robust quality assurance from manufacturers and the establishment of clear, species-specific regulatory standards to protect animal welfare and ensure predictable surgical outcomes [5].

## 5. Conclusions

The veterinary bone screws analyzed in this study exhibited non-uniform dimensions, indicating that they are either suboptimally manufactured or not produced to a consistently high level of precision. This variability may have implications for clinical outcomes, as inconsistent implant dimensions could affect mechanical stability and surgical reliability. The findings highlight the need for stricter quality control measures in veterinary implant production and underscore the potential value of establishing formal regulatory standards. Further research with a larger sample size, stratified by manufacturer, is warranted to validate these results and guide improvements in manufacturing precision.

## Figures and Tables

**Figure 1 vetsci-13-00038-f001:**
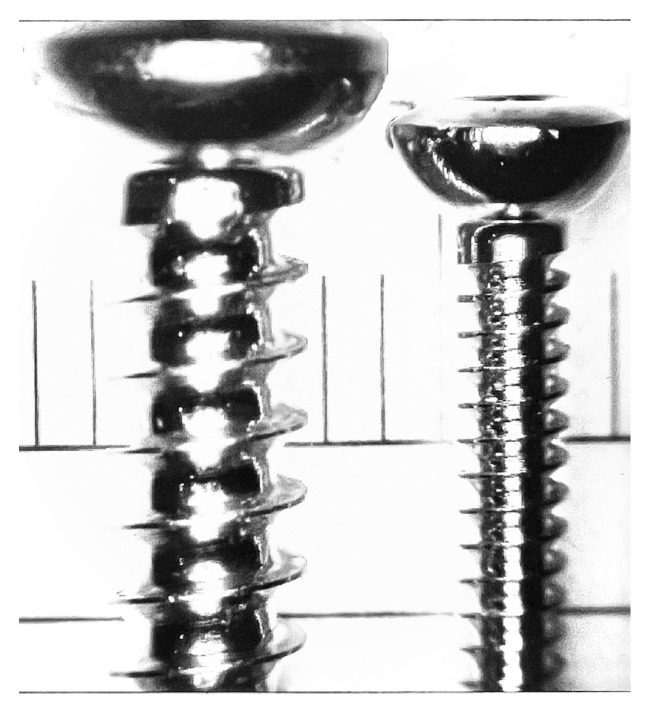
Representative threaded bodies of 3.5 mm and 2.0 mm cortical screws, shown with a transparent template. The graduated measurement markings on the template (labeled in millimeters, mm) served as a control reference for system calibration and accuracy verification prior to dimensional analysis.

**Figure 2 vetsci-13-00038-f002:**
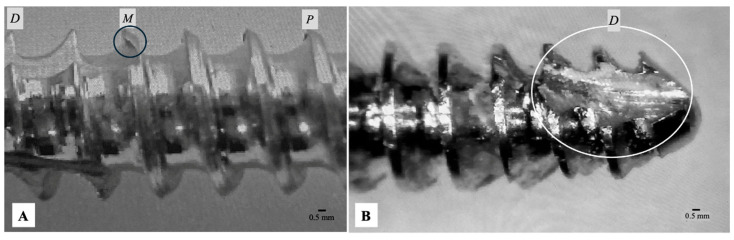
(**A**) Magnified view of a 2.0 mm cortical screw showing irregular thread form. Annotations (Proximal, (P), Middle (M), Distal (D) measurement regions) and highlighted area (black circle) where variations in major diameter and pitch were found outside tolerance, noting unevenness of thread crests and roots. (**B**) Microscopic view of a 3.5 mm cortical screw exhibiting a severe manufacturing defect. The visibly damaged thread section (white circle) at the distal (D) region demonstrates significant dimensional inaccuracy and compromised structural integrity, directly contributing to out-of-tolerance classification and potential mechanical compromise. A scale bar is included for reference.

**Table 1 vetsci-13-00038-t001:** Summary of Dimensional Analysis Results for Veterinary Bone Screws.

Screw Size	Parameter	Location	Compliance (n/N)	Compliance (%) (95% CI)	Mean (mm)	SD (mm)	CV%	Intra-Screw Variation (Out of Tolerance %)
**2.0 mm**	**Major Diameter**	**Overall**	**33/60**	**55% (42%, 68%)**	**2.029**	**0.052**	**2.57%**	**45% (27/60)**
		Proximal			2.029	0.052	2.57%	
		Middle			2.028	0.057	2.80%	
		Distal			2.030	0.046	2.27%	
	**Pitch**	**Overall**	**14/60**	**23% (13%, 36%)**	**0.488**	**0.045**	**9.22%**	**77% (46/60)**
		Proximal			0.490	0.046	9.40%	
		Middle			0.489	0.044	9.00%	
		Distal			0.485	0.045	9.29%	
**3.5 mm**	**Major Diameter**	**Overall**	**33/60**	**55% (42%, 68%)**	**3.571**	**0.104**	**2.91%**	**45% (27/60)**
		Proximal			3.560	0.106	2.98%	
		Middle			3.593	0.101	2.80%	
		Distal			3.561	0.104	2.91%	
	**Pitch**	**Overall**	**7/60**	**12% (5%, 23%)**	**1.186**	**0.146**	**12.35%**	**88% (53/60)**
		Proximal			1.182	0.147	12.44%	
		Middle			1.189	0.144	12.14%	
		Distal			1.186	0.146	12.35%	

**Table 2 vetsci-13-00038-t002:** Summary of Inferential Statistical Comparisons.

Comparison Type	Test Performed	Parameter	Comparison Groups	*p*-Value	Conclusion/Direction of Difference
**Proportion Compliant (Between Screw Sizes)**	Fisher’s Exact Test	Major Diameter	2.0 mm vs. 3.5 mm	0.999	No significant difference in major diameter compliance between screw sizes.
		Pitch	2.0 mm vs. 3.5 mm	0.045	2.0 mm screws show significantly higher pitch compliance than 3.5 mm screws.
**Intra-Screw Variability (Major Diameter vs. Pitch)**	*t*-test/Mann–Whitney U	Intra-Screw Range	2.0 mm: MD vs. Pitch	<0.001	For 2.0 mm screws, intra-screw variability (range) is significantly higher for pitch than for major diameter.
			3.5 mm: MD vs. Pitch	0.003	For 3.5 mm screws, intra-screw variability (range) is significantly higher for pitch than for major diameter.

## Data Availability

The original contributions presented in this study are included in the article/Appendix A. Further inquiries can be directed to the corresponding author.

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
