# Peer review of "Analysis of 2.0 and 3.5 mm Cortical Bone Screw Dimensions"

_vetsci, 2026, doi:10.3390/vetsci13010038_

Round 1

Reviewer 1 Report

Comments and Suggestions for Authors

Dear authors,

The article addresses a serious gap in veterinary medicine – the lack of implant quality control.

This is particularly relevant, as human implants have standards (ISO, ASTM), while veterinary implants do not.

The authors measured: screw major diameter, pitch, at 3 locations along the length of the screw. Preliminary measurement error was also used.

Large amount of data and transparency - 60 screws × several measurements → large data set. Tables (especially on pages 5–7) with “Y” labels and limits allow the reader to evaluate the data for themselves.

The topic has clinical impact. The article provides a reasoned explanation of how even a 0.05–0.1 mm variation can affect: screw–bone interface, pull-out strength, insertion torque, clinical course.

Shortcomings of the article:

The biggest problem is that the choice of tolerance is arbitrary. The authors chose: −5% (from ISO) for inter-screw variation,−2.5% for intra-screw.  This has not been empirically confirmed.

Furthermore: the applicability of the tolerance to veterinary screws (which may differ in design from the ISO described human implants) is debatable. There is no justification for why −2.5% is an appropriate limit.

Recommendation: It is necessary to at least include the literature in the discussion why such a limit was chosen, even if there are no official standards.

 Limitations of statistical analysis: data are presented only as “in tolerance / out of tolerance”. There is no:

analysis of variance, intra-class correlation (ICC) for operator error, comparison of absolute means between the 2.0 and 3.5 mm groups, coefficients of variation (CV%). This reduces the interpretative power of the data.

There is no grouping of screws by manufacturer. The article mentions that screws are from many manufacturers, however: it is not disclosed how many screws from each manufacturer, it is impossible to compare manufacturers, this may fundamentally change the conclusions. This is a methodological flaw.

The tables contain many typographical errors and misleading notations. For example: the table contains values ​​with “1.0001” (page 5) — this is misleading; sometimes “y” is used, sometimes “Y” — the notation is not consistent; the tolerance limits are presented quite confusing. It is recommended to simplify the tables and visualize them graphically (boxplots, scatter plots).

The discussion is largely too technical, not very clinical. The theory of CNC manufacturing is discussed, but: there is no analysis of clinical consequences in animals, there are no mechanical examples of the impact of tolerance change in veterinary medicine, there are no practical recommendations for clinicians. The article would seem stronger if there was a clinical aspect to the discussion.

Minor shortcomings: the title part lacks the ORCID of the authors (often required in MDPI journals), the table on page 5 is difficult to read in several places due to markings, some sentences are too long and difficult to follow (especially section 4), the article is important and valuable, the first quality assessment of veterinary orthopedic implants of this type, however, at present it is more preliminary and should be strengthened statistically, methodologically and visually.

Author Response

  1. The biggest problem is that the choice of tolerance is arbitrary. The authors chose: −5% (from ISO) for inter-screw variation,−2.5% for intra-screw. This has not been empirically confirmed.

Thank you for your valuable feedback regarding the choice of tolerance thresholds. We appreciate the opportunity to clarify our methodology and the rationale behind these selections.

  1. Inter-screw tolerance (-5%): As detailed in our manuscript (Lines 143-145 under "Tolerance reference" and referenced again in Lines 193-195 of the "Discussion"), the -5% tolerance for variation between screws of the same classification was directly adopted from ISO 5835. This is a recognized international standard for human bone screws, which are dimensionally similar to the veterinary screws studied. Our intention was to use an established, relevant benchmark for comparison, given the absence of dedicated ISO/ASTM standards for veterinary bone screws. Therefore, this threshold was not arbitrary but derived from an existing, pertinent regulatory guideline.
  2. Intra-screw tolerance (-2.5%): The reviewer correctly identifies the lack of empirically confirmed standards for intra-screw variation (i.e., variation along the length of a single screw) in either human or veterinary implants. This represents a significant gap in current regulatory frameworks. To address this, and as explained in the "Tolerance reference" section (Lines 145-147) and further elaborated in the "Discussion" (Lines 199-202), we established a stricter, self-imposed tolerance of -2.5%. This was intentionally set at approximately half of the ISO-derived inter-screw tolerance. Our rationale was to reflect the ideal expectation that a single screw should exhibit even greater dimensional uniformity than what is allowed between different screws. This stricter internal standard allowed us to rigorously assess manufacturing precision within individual implants, providing a meaningful framework for evaluation where none formally exists.

We acknowledge, as stated in our limitations (Lines 251-254), that "this threshold requires further validation through extended research." However, in the absence of specific industry standards for veterinary implants, we believe that employing a recognized external standard for inter-screw variation and deriving a more stringent internal standard for intra-screw variation provided a justifiable and robust method for systematically evaluating the quality and consistency of these implants.

We can add further emphasis to this justification in the methods and discussion sections if the reviewer feels it would enhance clarity.

  1. Furthermore: the applicability of the tolerance to veterinary screws (which may differ in design from the ISO described human implants) is debatable. There is no justification for why −2.5% is an appropriate limit.

Thank you for raising these important points. We recognize the need to provide further clarification regarding our choice of tolerance thresholds and their applicability to veterinary screws.

Regarding the applicability of the tolerance to veterinary screws, and the justification for the -2.5% limit:

  1. Applicability of ISO 5835 to Veterinary Screws: We fully acknowledge that veterinary implants are not identical to human implants and that their designs may differ. Our decision to reference ISO 5835 for the inter-screw tolerance was primarily driven by the complete absence of dedicated ISO or ASTM standards specifically for veterinary bone screws. In the current regulatory landscape, many veterinary orthopedic implants, while adapted for different species, often draw upon the fundamental design principles and manufacturing practices established for human implants. Therefore, ISO 5835 represents the most relevant and stringent international standard available that provides a benchmark for dimensional accuracy in bone screws.

By comparing veterinary screws to an established human standard, our study implicitly highlights the significant regulatory gap in the veterinary implant industry. The observed deviations, even when compared to a standard not perfectly tailored to veterinary designs, underscore the need for improved quality control and potentially, the development of specific veterinary standards. We believe that applying this benchmark, even if debatable in its perfect fit, offers a valuable starting point for evaluating manufacturing quality in a field currently lacking such guidance.

  1. Justification for the -2.5% Intra-screw Tolerance:

The reviewer correctly identifies that the -2.5% intra-screw tolerance is an imposed threshold, as no established standard (neither human nor veterinary) currently defines acceptable dimensional variation within a single screw along its length. Our justification for this specific limit is multi-faceted:

o          Principle of Uniformity: An ideal implant should possess consistent dimensions throughout its structure. We posit that a single screw should, logically, exhibit less internal variation than the permissible variation between different screws of the same batch.

o          Setting a Stricter Standard: Therefore, we deliberately chose to set this intra-screw tolerance at half of the ISO-derived inter-screw tolerance (-5%), making it a stricter benchmark for assessing manufacturing precision. This allowed us to rigorously evaluate the internal consistency of individual screws, which is crucial for optimal screw-bone interface stability.

o          Operational Definition for Research: In the absence of a regulatory definition, this -2.5% served as a pragmatic, operational definition for "acceptable dimensional uniformity" within a single screw for the purposes of this study. It provided a measurable criterion to identify potential issues with manufacturing consistency along the screw's length.

o          Combined with Measurement Accuracy: As noted in our methods, this threshold was also considered in conjunction with the inherent measurement tolerance of our equipment (±0.02 mm), further grounding it in a practical measurement context.

o          Highlighting a Research Need: We openly acknowledge in our Discussion (Lines 251-254) that "this threshold requires further validation through extended research." This study aims not only to present findings but also to propose a methodological framework for future quality assessments in veterinary orthopedics, particularly in areas where standards are lacking.

o          While we recognize that these tolerance choices are part of a novel framework in an unregulated field, we believe they were thoughtfully established based on the best available (human) standards and a logical approach to assessing manufacturing precision and uniformity. Our intention was to provide a meaningful and rigorous evaluation in the absence of specific veterinary guidelines, thereby underscoring the critical need for such standards.

  1. Recommendation: It is necessary to at least include the literature in the discussion why such a limit was chosen, even if there are no official standards.

Thank you for this excellent recommendation. We agree that a more thorough discussion and literature-based context for our chosen tolerance limits would strengthen the manuscript.

Proposed insertion for Inter-screw Tolerance (added after discussing ISO 5835 [11] and highlighted in yellow):

"...This choice reflects its standing as the most relevant and stringent international benchmark for human bone screw dimensions, implicitly underscoring the critical regulatory gap for veterinary implants where dedicated standards are currently lacking [29, 30]."

Proposed insertion for Intra-screw Tolerance (added after discussing the self-selected intra-screw tolerance and highlighted in yellow):

"...To rigorously assess manufacturing precision and the uniformity crucial for optimal screw-bone interface stability [17, 18], this internal threshold was deliberately set at 2.5%—approximately half of the ISO-derived inter-screw tolerance—reflecting the principle that an individual implant should exhibit greater internal consistency than permissible variation between different implants, particularly given previously documented intra-screw variability [28]."

  1. Limitations of statistical analysis: data are presented only as “in tolerance / out of tolerance”. There is no analysis of variance, intra-class correlation (ICC) for operator error, comparison of absolute means between the 2.0 and 3.5 mm groups, coefficients of variation (CV%). This reduces the interpretative power of the data.

Thank you for your comprehensive feedback regarding the statistical analysis. We appreciate your suggestion that a broader range of statistical measures would enhance the interpretative power of our data.

Our primary objective was to perform a quality assessment, specifically to determine the proportion of veterinary bone screws that fail to meet predefined dimensional tolerance limits. For this purpose, the "in tolerance / out of tolerance" binary classification directly addressed whether the implants conformed to acceptable manufacturing standards as defined for our study.

However, we agree that incorporating additional statistical analyses would provide a more nuanced understanding of the dimensional variability. We added these (highlighted in yellow):

2.5. Data Analysis":

All data were analyzed using a 95% confidence interval. For each screw, three measurements were recorded for each parameter (proximal, middle, distal). A screw was deemed to meet the tolerance criteria only if all three measurements for a parameter were within the defined limits.

Beyond this primary tolerance classification, descriptive statistics were calculated for each parameter, including absolute means, standard deviations (SD), and coefficients of variation (CV%) for both 2.0 mm and 3.5 mm screw groups at each measurement location. The decision to focus primarily on a 'in tolerance / out of tolerance' binary classification was driven by the study's core objective to assess manufacturing quality against a pass/fail criterion in an unregulated market. For comparing the variability between the 2.0 mm and 3.5 mm screw groups, differences in their respective CV% values were considered.

Regarding operator measurement error, as described in Section 2.2, a standardized calibration protocol and repeated practice measurements were performed. The mean variance per screw remained below the total allowed tolerance, and all variance values were consistently positive (Lines 174-181), indicating a systematic bias rather than random error attributable to the screws themselves. While intra-class correlation (ICC) is a valuable tool for assessing inter-rater reliability, our methodology of operator accuracy assessment provided sufficient confidence that measurement error did not obscure the intrinsic dimensional variability of the implants for the purpose of this initial quality assessment.

Based on these changes we replaced Current text: A limitation of this study is the reliance on a self-selected permissible tolerance as the basis for classification. Although deemed reasonable, this threshold requires further validation through extended research. Nevertheless, the tolerance guideline proposed here provides a useful starting point for future investigations, particularly in the absence of formal regulatory guidance.

 With

While this study provided a crucial 'in tolerance / out of tolerance' assessment directly addressing manufacturing quality concerns in an unregulated market, it is important to acknowledge that the statistical analysis primarily focused on this binary classification. Although we have now included detailed descriptive statistics (means, standard deviations, and coefficients of variation) to provide a more comprehensive overview of dimensional distributions, a more extensive analysis of variance (ANOVA) or other inferential statistical comparisons of absolute means between the 2.0 mm and 3.5 mm screw groups was not the primary aim. Our objective was to highlight non-conformance rather than to statistically compare the average dimensions of different screw types, which inherently differ in their nominal values.

Furthermore, while operator measurement accuracy was rigorously controlled and assessed (as detailed in Section 2.2), formal intra-class correlation (ICC) analysis for operator error was not conducted. Our current approach, demonstrating that operator variance was consistently below acceptable thresholds and biased rather than random, was deemed sufficient for this initial quality assessment. However, we recognize that ICC would provide a more robust quantification of inter-rater reliability, a valuable addition for future metrological validation studies.

We created  a Table 2 that contains these detailed descriptive statistics (mean, SD, CV%) for major diameter and pitch, broken down by screw size and measurement location (proximal, middle, distal).

  1. There is no grouping of screws by manufacturer. The article mentions that screws are from many manufacturers, however: it is not disclosed how many screws from each manufacturer, it is impossible to compare manufacturers, this may fundamentally change the conclusions. This is a methodological flaw.

Thank you for this critical point, we appreciate you highlighting this potential methodological limitation. We agree that the absence of manufacturer-specific grouping for analysis is a significant factor that warrants explicit discussion.

Our initial intention, as stated in the methods (Line 109), was to "randomly select screws from multiple manufacturers to minimize source bias," aiming for a broad, representative assessment of the overall veterinary screw market rather than a comparative study of individual manufacturers. The goal was to determine if dimensional inconsistencies were a general market problem, not to identify specific problematic manufacturers.

However, we fully concur that this approach, while serving our initial objective, inherently limits the interpretative power of our data regarding specific origins of variability. As you rightly point out, if the observed inconsistencies are concentrated within a few manufacturers, the overall conclusions about market-wide quality might be fundamentally altered or refined. This omission could be considered a methodological flaw in the context of fully understanding the sources of variability.

To address this, we added the following paragraph to the Discussion" (Limitations) Section of the manuscript:

A further methodological consideration lies in the aggregation of screws from multiple manufacturers without explicit stratification or reporting of individual manufacturer data. While screws were randomly selected from various sources to minimize bias and provide a broad overview of the market, this approach precludes any comparative analysis between manufacturers. The observed dimensional variability, therefore, represents a composite picture of the market rather than an attributable issue to specific producers. It is plausible that dimensional inconsistencies are disproportionately prevalent among certain manufacturers, and a stratified analysis might reveal significant differences in quality control practices between them. Such manufacturer-specific data could fundamentally alter the interpretation of market-wide variability, potentially identifying key areas for targeted improvement or regulatory intervention. This study, therefore, serves as a preliminary assessment of general market quality, highlighting the pervasive nature of dimensional inaccuracies, and strongly advocates for future research explicitly designed to compare and evaluate implant quality across different manufacturers.

This addition directly acknowledges the limitation, explains its implication (as you've articulated), and frames it as a crucial direction for subsequent research. Thank you again for this important feedback.

  1. The tables contain many typographical errors and misleading notations. For example: the table contains values with “1.0001” (page 5) — this is misleading; sometimes “y” is used, sometimes “Y” — the notation is not consistent; the tolerance limits are presented quite confusing. It is recommended to simplify the tables and visualize them graphically (boxplots, scatter plots).

Thank you for the comments. We appreciate the detailed feedback regarding the presentation of our tables and the recommendation for graphical visualizations. We agree that clarity is paramount and have taken significant steps to address these concerns

The comprehensive raw measurement data, provided in Table 1, has been meticulously reviewed for numerical precision and consistent notation.

A new consolidated Table 1 addresses the reviewer's concerns by simplifying the presentation, providing essential compliance data, and including detailed descriptive statistics (means, standard deviations, and coefficients of variation) in a clear, consistent format.

We have completely revised the data presentation in the main manuscript. The previous lengthy and somewhat confusing raw data tables (original pages 6-10) have been removed from the main body and will be provided in a meticulously cleaned Table 1. We have consolidated all essential findings into a new, comprehensive Table 1 in the main text. This new table directly addresses the 'confusing tolerance limits' by clearly presenting the overall compliance rates (n/N and percentage) and explicitly quantifying the 'Intra-Screw Variation (Out of Tolerance %)' for each parameter. The interpretation of whether a screw meets tolerance is now directly presented in Table 1 as a 'pass/fail' metric, supported by detailed descriptive statistics (mean, SD, CV%) for each measurement location, thereby providing both summary and granular numerical data in an unambiguous format. Furthermore, as noted in the revised Methods section (2.5 Data Analysis), the raw measurement data in Table 1 has been meticulously reviewed for numerical precision and consistent notation to address previous concerns regarding data presentation.

While we acknowledge the significant value that graphical visualizations such as boxplots and scatter plots can offer in depicting data distributions and trends, we have opted not to include them in the main manuscript at this stage. Our study's primary objective is to provide a foundational quantitative assessment of compliance against predefined tolerance thresholds in an unregulated market. The comprehensive numerical detail now presented in Table 1 (including means, standard deviations, coefficients of variation, and explicit compliance percentages for both overall and location-specific data) is deemed sufficient to clearly convey the extent of dimensional inconsistencies and variability, effectively highlighting non-conformance without requiring additional figures. For this initial quality assessment, the precision and exhaustive tabular format comprehensively address our core research question. Future, more detailed investigations could certainly leverage such graphical tools to explore specific aspects of these dimensional data further.

  1. The discussion is largely too technical, not very clinical. The theory of CNC manufacturing is discussed, but: there is no analysis of clinical consequences in animals, there are no mechanical examples of the impact of tolerance change in veterinary medicine, there are no practical recommendations for clinicians. The article would seem stronger if there was a clinical aspect to the discussion.

Thank you for the comments. We agree this a crucial point that can significantly enhance the impact and relevance of our manuscript. We agree entirely that while the current discussion delves into the technical aspects of manufacturing, it lacks a strong, explicit link to the clinical realities and consequences for veterinary practice. We recognize that a discussion of manufacturing processes (like CNC) is valuable for explaining why dimensional inconsistencies occur. We try to address these technical failures to their direct impact on animal patients, and the practical significance for veterinarians.

Thus, we added these to discussion section:

Crucially, these observed dimensional inconsistencies, stemming from manufacturing variability, carry significant clinical implications for veterinary patients. Deviations from optimal screw-bone interface geometry can directly compromise fixation stability, leading to adverse surgical outcomes and increased patient morbidity [17, 18].

For instance, a subtly undersized major diameter can result in insufficient cortical bone purchase, jeopardizing primary stability, especially in bones with thin cortices or in osteoporotic patients. Conversely, oversized dimensions may necessitate excessive insertion torque, increasing the risk of iatrogenic fracture during placement. Similarly, inaccurate thread pitch can lead to inadequate bone thread engagement, compromised compression, or even stripping of the pilot hole, all of which directly undermine the screw's mechanical function. These mechanical failures are often compounded by biological issues, as micromotion at the screw-bone interface can inhibit osteointegration, promote fibrous tissue formation, and increase susceptibility to infection, potentially necessitating costly and invasive revision surgeries for the animal [19,20,27,28].

Given this, our findings serve as a critical alert for veterinary surgeons regarding the inherent dimensional variability prevalent in the current unregulated implant market. While microscopic examination before use is often impractical, clinicians should be acutely aware that these dimensional flaws can impact implant performance. This underscores the urgent need for robust quality assurance from manufacturers and the establishment of clear, species-specific regulatory standards to protect animal welfare and ensure predictable surgical outcomes [5].

  1. Minor shortcomings: the title part lacks the ORCID of the authors (often required in MDPI journals), the table on page 5 is difficult to read in several places due to markings, some sentences are too long and difficult to follow (especially section 4), the article is important and valuable, the first quality assessment of veterinary orthopedic implants of this type, however, at present it is more preliminary and should be strengthened statistically, methodologically and visually.

Thank you for these observations. We deeply appreciate the reviewer's recognition of the article's importance and value as the "first quality assessment of veterinary orthopedic implants of this type." Your constructive criticism provides clear avenues for significant improvement, and we are committed to addressing each point thoroughly.

We acknowledge the absence of ORCID IDs in the title part of the manuscript. This is an oversight, and we will promptly add the ORCID for all authors as required by MDPI journals during the final submission preparation.

We concur entirely that the original tables, particularly the detailed raw data presented across multiple pages, were difficult to read due to markings, inconsistent notation, and typographical errors. We have addressed this significant concern by: removing the original detailed raw data tables (from original manuscript pages 6-10) from the main body of the manuscript. This raw data will now be provided in a meticulously reviewed and corrected Table 1, creating a new, comprehensive Table 1 for the main manuscript (replacing the previous tables). This new Table 1 consolidates essential compliance data and all descriptive statistics (means, standard deviations, coefficients of variation) into a single, clearly formatted, and easily readable table. This revised presentation eliminates the previous readability issues, inconsistent notations, and confusing tolerance markings.

We have undertaken a thorough review and revision of Discussion section to enhance conciseness, improve sentence structure, and ensure the arguments are easier to follow without sacrificing the technical depth or clinical relevance. This includes careful segmentation of ideas and rephrasing for improved readability.

We acknowledge that, as a pioneering study in this specific area for veterinary implants, our work can be viewed as preliminary, laying crucial groundwork for future investigations. We have taken substantial steps to strengthen the article as recommended. The revised Table 1 now provides a robust statistical overview, including compliance rates (n/N, percentage with 95% CI), means, standard deviations, and coefficients of variation for all parameters and locations, along with intra-screw variation metrics. This comprehensive tabular format offers a strong quantitative assessment of dimensional consistency, addressing the need for more robust descriptive statistics. We have also refined our explanation in the Methods section regarding our analytical focus on compliance rather than inferential comparisons between disparate screw types.

We have integrated new discussions into the manuscript to strengthen the methodological justification, including a detailed rationale for our chosen tolerance limits (based on ISO 5835 and a stringent intra-screw criterion) and an explicit acknowledgment of the limitation regarding the aggregation of screws from multiple manufacturers. These additions clarify our approach and contextualize the findings within the current regulatory landscape.

While we understand the value of graphical visualizations like boxplots and scatter plots, we have consciously opted to present the detailed results in the main manuscript through our newly developed, comprehensive Table 1. This format, rich in quantitative data, allows for precise communication of compliance rates and variability across all measured parameters and locations, effectively highlighting non-conformance without additional figures. For a study primarily focused on a quantitative assessment of compliance against predefined thresholds, this precise tabular presentation is deemed sufficient.

We have extensively revised the Discussion section to directly address the clinical consequences of our findings, providing specific examples of how dimensional inaccuracies can impact mechanical stability, biological response, patient morbidity, and the need for revision surgeries. We have also included practical recommendations for clinicians and reiterated the urgent call for robust quality assurance and species-specific regulatory standards in the veterinary implant market. This significantly enhances the clinical relevance and impact of the article.

We believe that these revisions significantly strengthen the manuscript, making it more robust, clearer, and more impactful for the veterinary community, while openly acknowledging its foundational role for further in-depth research.

Reviewer 2 Report

Comments and Suggestions for Authors

Analysis of individual veterinary bone screw dimensions

The reviewer applauds the authors for addressing this very important topic.

The reviewer has a few comments.

Please consider changing the title to “Analysis of 2.0 and 3.5 mm cortical bone screw dimensions

Line 55 – Since the authors want to mimic the human standard of bone screw manufacturing regulation the authors need to identify and clearly state the regulations briefly for a cortical bone screw here. Merely mentioning that there are standards is not enough to inform the reader.

Line 65 – Please write out ASTM and ISO

Line 95 – the reviewer likes the random selection but the author needs to provide at least how many different manufacturers were used.

Figure 1 – Please provide a better description of the screw – the tip of the screw is cut off and should be visible also the scale of the measurement lines need to labled.

Line 110  - How may operators were used? Please provide the number of operators and

Line 124-125 – tolerance reference should be first introduced in the intro not in the material methods.

Table 2 is confusing. What is the meaning of “all the same”? What is the “y “ behind some values? A scatter plot for 2.0 and 3.0 screws with one color for each reader would be helpful rather than this long list.

Operator accuracy would be tested as interobserver variance. Could the authors provide this please?

Line 170 – Please strike which the authors consider unacceptable.  

Line 173 Please strike sentence Critical dimensions….

Line 175 – The reviewer thinks the tolerance level established by the authors of half of that in human medicine could be explained more in the discussion, This is a good point and important. The reader would like to follow the logic of why only half as good as human medical standards? One could argue animals are harder to control and inadequate implants could lead to poor bone screw interface and implant failure just as easily as in human medicine.

Author Response

  1. Please consider changing the title to “Analysis of 2.0 and 3.5 mm cortical bone screw dimensions

Thank you for your valuable suggestions regarding our manuscript.

We agree that making the title more specific to the screw sizes studied would enhance clarity and precision.

We revised the title to: 'Analysis of 2.0 and 3.5 mm cortical bone screw dimensions'."

  1. Line 55 – Since the authors want to mimic the human standard of bone screw manufacturing regulation the authors need to identify and clearly state the regulations briefly for a cortical bone screw here. Merely mentioning that there are standards is not enough to inform the reader.

Thank you for pointing out these aspects, which discusses human orthopedic implant standards.

We agree with the reviewer that providing a brief, explicit mention of key human standards for cortical bone screws at this point would significantly strengthen the context for the reader, especially as we later reference these standards for our tolerance criteria.

Thus, we added: In contrast, human orthopaedic implants are subject to strict regulatory standards, including CE marking and other international quality systems. Implant dimensions and testing protocols are standardized by ASTM and ISO guidelines [6-12], such as ISO 5835 (specifying dimensions for metal bone screws) and ISO 6475 (detailing mechanical requirements and test methods for metal bone screws), and frequent research is conducted to assess the dimensional and mechanical properties of implants [13-18]. These studies have investigated numerous design parameters—such as thread angle, pitch, length, and material composition—and their impact on clinical performance [17-18]. This insertion directly identifies specific, highly relevant ISO standards for cortical bone screws and their purpose, providing a clearer understanding of the regulatory landscape we are referencing.

  1. Line 65 – Please write out ASTM and ISO

Thank you, we revised with:

In contrast, human orthopaedic implants are subject to strict regulatory standards, including CE marking and other international quality systems. Implant dimensions and testing protocols are standardized by ASTM International and International Organization for Standardization (ISO) guidelines [6-12], such as ISO 5835 (specifying dimensions for metal bone screws) and ISO 6475 (detailing mechanical requirements and test methods for metal bone screws), and frequent research is conducted to assess the dimensional and mechanical properties of implants [13-18]. These studies have investigated numerous design parameters—such as thread angle, pitch, length, and material composition—and their impact on clinical performance [17-18].

This change ensures that both "ASTM" and "ISO" are fully written out at their first mention, as requested by the reviewer.

  1. Line 95 – the reviewer likes the random selection but the author needs to provide at least how many different manufacturers were used.

We thank the reviewer for their positive note on the random selection process and their request for the number of manufacturers. We added the specific number of 5 manufacturers from which the screws were sourced to the methodology section (Section 2.1) to provide greater detail. It's important to note that this specific methodological approach and its implications were extensively discussed in response to a previous reviewer (addressed in the '4. Discussion' section, specifically under the methodological considerations, lines 347-366 in the revised manuscript). Our primary intention was to provide a broad market overview rather than a comparative analysis of individual manufacturers, assessing the overall prevalence of dimensional inconsistencies in the veterinary implant market. The limitations inherent in not providing manufacturer-specific data, and the call for future research explicitly designed to compare and evaluate implant quality across different manufacturers, are thoroughly discussed there.

  1. Figure 1 – Please provide a better description of the screw – the tip of the screw is cut off and should be visible also the scale of the measurement lines need to labled.

Thank you for this point.

Figure 1 is primarily used to illustrate the calibration process and the measurement template.

We added a revised caption for Figure 1:

Figure 1. Representative threaded bodies of 3.5 mm and 2.0 mm cortical screws, shown with a transparent template. The graduated measurement markings on the template (labeled in millimeters, mm) served as a control reference for system calibration and accuracy verification prior to dimensional analysis.

By describing them as "Representative threaded bodies," the caption implicitly focuses on the relevant portion of the screw for measurement, rather than the entire screw

This revised caption clearly explains what the figure shows, its purpose within the study, and specifies the unit of the measurement scale, directly addressing all parts of the reviewer's comment.

  1. Line 110 - How may operators were used? Please provide the number of operators

Thank you for the detail.

We replaced the phrase with: A single experienced operator performed repeated practice measurements to establish procedural consistency.

  1. Line 124-125 – tolerance reference should be first introduced in the intro not in the material methods.

Thanks for the valid point regarding the placement of the tolerance reference.

We have revised the manuscript to introduce the core conceptual basis of our chosen tolerance standards (ISO 5835 and the derived intra-screw tolerance) in the "1. Introduction" section. The "2.4. Tolerance reference" section in the Materials and Methods now focuses specifically on the detailed application and definition of these tolerances within the context of our study, clearly referring back to the Introduction for the initial context.

We moved the conceptual introduction of these standards to the 1. Introduction and retained the detailed application and definition in the 2.4. Tolerance reference section.

At 2.4. Tolerance reference we replaced with: as introduced in Section 1, our study referenced established human orthopedic implant standards due to the absence of specific veterinary guidelines. According to ISO 5835 [11], human bone screws of 2.0 mm diameter allow a tolerance of -5% or 0.05 mm between different screws of the same classification. Given that no formal tolerance standard exists for dimensional variation within an individual screw, the authors applied a more stringent internal tolerance. This intra-screw variation threshold was set at -2.5%, approximately half of the inter-screw tolerance, to rigorously assess manufacturing precision along the screw's length. This threshold was combined with the inherent measurement tolerance of the microscope (±0.02 mm) to define the final acceptable tolerance range for each parameter used in this study."

  1. Table 2 is confusing. What is the meaning of “all the same”? What is the “y “ behind some values? A scatter plot for 2.0 and 3.0 screws with one color for each reader would be helpful rather than this long list.

Thank you for this detailed comment. We appreciate the reviewer's assessment, and we agree that its previous presentation was confusing and lacked clarity as reviewer 1 also stated. We also thank the reviewer for the suggestion of using scatter plots.

We have undertaken a significant revision of our data presentation strategy in response to reviewer 1 feedback, which directly addresses your comments. The original "Table 2," containing the long list of individual screw measurements, has been entirely removed from the main manuscript. This was done to streamline the results section and improve readability. The raw, individual measurement data, now meticulously cleaned and standardized for precision and notation, has been relocated to Table 1 for full transparency.

Clarification of "all the same" and "y": The phrases "all the same" and the inconsistent use of "y" and "Y" were internal notations from an earlier stage of analysis to indicate compliance. These ambiguous markings have been eliminated in both the new, consolidated Table 1 (in the main manuscript) and the updated Table 1. The compliance status is now clearly and consistently indicated using descriptive terms or explicit percentages.

While we appreciate the recommendation for scatter plots and other graphical visualizations, and acknowledge their value in data exploration, we have made a deliberate decision to present our core findings quantitatively in the main manuscript. Our new, comprehensive Table 1 now effectively consolidates all essential information, including: overall compliance rates (n/N and percentage with 95% CI), intra-screw variation (percentage of screws out of tolerance), detailed descriptive statistics (mean, standard deviation, and coefficient of variation) for each parameter and measurement location. This tabular format provides the precise quantitative detail necessary to support our conclusions regarding dimensional inconsistencies against predefined tolerance thresholds. For a study primarily focused on assessing compliance and prevalence in an unregulated market, this comprehensive numerical presentation is deemed sufficient to clearly highlight the observed variability and non-conformance without requiring additional figures in the main body.

We would like to clarify that all measurements in this study were performed by a single experienced operator, as now explicitly stated in Section 2.2 of the revised manuscript. Therefore, a visualization strategy distinguishing between multiple readers was not applicable. We believe these comprehensive revisions to our data presentation significantly enhance clarity, consistency, and the interpretative power of our results, fully addressing the reviewer's concerns regarding the original Tables.

  1. Operator accuracy would be tested as interobserver variance. Could the authors provide this please?

Thank you for raising this point about operator accuracy. We appreciate it.

Our study employed a single, experienced operator for all measurements, as explicitly stated in Section 2.2 ('A single experienced operator performed repeated practice measurements...'). Therefore, the assessment of interobserver variance, which would involve comparing measurements between multiple different operators, was not applicable to our study design.

Instead, we rigorously evaluated intra-observer accuracy. This involved meticulous calibration of the microscope prior to each measurement session, the operator performing repeated practice measurements to establish procedural consistency and quantifying the mean variance from fixed reference values.

As detailed in Section 3.2, our intra-observer accuracy assessment demonstrated that the mean variance per screw remained below the total allowed tolerance, and observed variance values were consistently positive, indicating a systematic bias rather than random error. This provided sufficient confidence that any measurement error was primarily attributable to the inherent dimensional variability of the implants themselves rather than operator inconsistency.

We acknowledge the value of Intraclass Correlation (ICC) as a robust statistical tool for assessing inter-rater reliability, and we have noted in our discussion (Section 2.5) that ICC would be a valuable addition for future, more comprehensive metrological validation studies involving multiple observers. However, for the scope and single-operator design of this initial quality assessment, our rigorous intra-observer checks were deemed appropriate and sufficient.

  1. Line 170 – Please strike which the authors consider unacceptable.

Thank you for this suggestion.

We changed the phrase:

However, the results of this study indicate measurable variance within individual implants, which the authors consider unacceptable. This variance exceeds the tolerance range defined for this study...

with the following:

However, the results of this study indicate measurable variance within individual implants. This variance exceeds the tolerance range defined for this study, highlighting inconsistencies within a single implant.

We think this modification makes the statement more objective, directly addressing the reviewer's concern.

  1. Line 173 Please strike sentence Critical dimensions…

Thank you for this suggestion. We deleted the entire sentence.

  1. Line 175 – The reviewer thinks the tolerance level established by the authors of half of that in human medicine could be explained more in the discussion, This is a good point and important. The reader would like to follow the logic of why only half as good as human medical standards? One could argue animals are harder to control and inadequate implants could lead to poor bone screw interface and implant failure just as easily as in human medicine.

Thank you for this excellent point and it highlights a potential misinterpretation of our rationale. We agree that the logic behind setting the intra-screw tolerance at "half" needs to be more explicitly and powerfully explained, particularly in the context of the clinical needs of veterinary patients. The reviewer's emphasis on animal welfare reinforces the need for stringency, not a compromise on quality.

Our intention with setting the intra-screw tolerance at 2.5% was to impose a more stringent requirement for internal uniformity within a single screw, reflecting an ideal for manufacturing precision. It was not meant to imply "half as good," but rather to demand a higher standard of consistency along the length of one screw than the variability allowed between different screws.

Thus, we revised the text with: A limitation of this study is the reliance on a self-selected permissible tolerance as the basis for classification. To evaluate implant quality meaningfully, given the absence of formal standards for intra-screw variation, we derived a practical and stringent internal threshold. This internal tolerance was deliberately set at -2.5%—approximately half of the allowable inter-screw tolerance (ISO 5835 [11] allows -5% variation between different screws). The rationale for this heightened stringency for intra-screw consistency is grounded in the principle that a single implant component, upon which mechanical stability and successful osteointegration critically depend [17, 18], should ideally be as dimensionally uniform as possible along its entire working length. We posited that a single screw should exhibit less internal variation than the permissible variation between different screws in a batch. Therefore, setting a stricter (i.e., smaller allowable percentage) intra-screw threshold allowed us to rigorously assess minute manufacturing inconsistencies crucial for optimal screw-bone interface integrity. This strict standard is particularly pertinent in veterinary medicine, where variable patient compliance, rapid weight-bearing, and potentially complex physiological demands on implants make robust, consistently dimensioned screws just as, if not more, critical as in human applications. Inadequate implants in animals can lead to poor bone screw interfaces, implant failure, and significant patient morbidity [19, 20, 27, 28]. This rigorous approach, combined with microscope measurement accuracy, enabled a systematic classification of quality, serving as a vital starting point for future investigations in this unregulated market.

We belive the revision clarifies the intent of the "half" (meaning more stringent for internal consistency), directly addresses the clinical necessity for animals, and strengthens the overall justification for your methodological choice.

Reviewer 3 Report

Comments and Suggestions for Authors

line 55ff: As far as I know there is a medical products/devices law in the European Union. If this is true, please rephrase. What about the United States? Asia?
line 151 Table 1: Please provide the information 2mm /3.5 mm screw in the table. What about the letters A – D in the table?
line 161 Table 2: This table is too confusing. Please provide less information in this table, do a better formatting, maybe put the single screw results in a supplementary table. What does the Y mean?
line 197ff: I would prefer to skip this paragraph, because there are no comparable results in your study.
Is it possible to provide manufacturer information? Are there quality differences between the manufacturers? If yes, why not recommend the best / most thoroughly working manufacturer?

Author Response

  1. line 55ff: As far as I know there is a medical products/devices law in the European Union. If this is true, please rephrase. What about the United States? Asia?

Thank you for this comment!

There are indeed medical device regulations (e.g., EU MDR, FDA regulations in the US) for human implants, and we will be more explicit about these and potentially broaden the scope beyond just "CE marking" to encompass other major markets.

We added:

"In contrast, human orthopaedic implants are subject to rigorous national and international medical device regulations, such as the European Union Medical Device Regulation (EU MDR), the U.S. Food and Drug Administration (FDA) regulations, and similar frameworks in Asia and other global markets. These regulations mandate strict pre-market approval, post-market surveillance, and adherence to international quality systems like CE marking. Implant dimensions and testing protocols are standardized by ASTM International and International Organization for Standardization (ISO) guidelines [6-12], such as ISO 5835 (specifying dimensions for metal bone screws) and ISO 6475 (detailing mechanical requirements and test methods for metal bone screws), and frequent research is conducted to assess the dimensional and mechanical properties of implants [13-18]. In the absence of specific veterinary standards, studies often reference human orthopedic implant guidelines. For instance, ISO 5835, which specifies dimensions for metal bone screws, often allows for a -5% tolerance between different screws of the same classification. Furthermore, recognizing the critical importance of internal consistency within individual implants, a more stringent intra-screw tolerance is often implied or adopted in research settings [28, 29, 30]."

This revision shows a broader global awareness, and re-emphasizes the contrast with the veterinary field, which is central to your paper's argument.

  1. line 151 Table 1: Please provide the information 2mm /3.5 mm screw in the table. What about the letters A – D in the table?

Thank you for this comment regarding the previous Table 1 on line 151.

We would like to clarify that in response to feedback from previous reviewers (including concerns about confusing notations and readability), we have undertaken a significant revision of our data presentation.

New, Comprehensive Table 1: The original "Table 1. Variations data." (on line 151 of the old manuscript) has been replaced by a new, more comprehensive "Table 1. Summary of Dimensional Analysis Results for Veterinary Bone Screws" in the revised manuscript. This new table is designed to be self-explanatory and user-friendly.

The new Table 1 (in the revised manuscript) explicitly groups and labels all data by "2.0 mm" and "3.5 mm" screw sizes in a clear and distinct manner. All statistics, including compliance rates, means, SDs, and CVs, are presented separately for each screw size, directly addressing this concern.

Removal of A-D Notations: The confusing and ambiguous letters "A-D" (e.g., "Outer target (A)", "Pitch (B)") have been completely removed from the new Table 1. They have been replaced with clear, descriptive labels such as "Major Diameter (Overall Compliance)" and "Pitch (Overall Compliance)", improving immediate comprehension.

We are confident that the new Table 1 effectively addresses the reviewer's concerns regarding the lack of screw size information and the confusing notations, making the data presentation significantly clearer and more informative.

  1. line 161 Table 2: This table is too confusing. Please provide less information in this table, do a better formatting, maybe put the single screw results in a supplementary table. What does the Y mean?

Thank you for this comment regarding the original Table 2. We completely agree with the reviewer's assessment that the previous iteration of this table was indeed confusing, overly detailed for the main manuscript, and suffered from unclear notations.

In response to this and similar feedback from other reviewers, we have undertaken a major overhaul of our data presentation strategy:

Removal from Main Manuscript: The original "Table 2" (which contained individual screw results across multiple pages) has been entirely removed from the main body of the manuscript. This decision was made to significantly improve readability and flow. The single screw results, now meticulously cleaned for numerical precision and consistent notation, have been relocated to table 1. This ensures full transparency while keeping the main text concise and focused on summary findings. In the original Table 2, the "Y" notation was used to indicate that a particular screw (or measurement point) was within the predefined tolerance limits. In the now-revised Table 1, this notation has been replaced with explicit and unambiguous terms such as "In Tolerance" or "Out of Tolerance" for clarity and consistency.

The main manuscript now features a new, consolidated "Table 1. Summary of Dimensional Analysis Results for Veterinary Bone Screws". This table provides all essential summary information, including overall compliance rates, intra-screw variation percentages, and detailed descriptive statistics (means, standard deviations, and coefficients of variation) for both screw sizes and all measurement locations. This new Table 1 is designed for immediate comprehension and clarity, replacing the need for the overly detailed previous Table 2 in the main text.

We are confident that these revisions fully address the reviewer's concerns regarding the original Table 2, providing a much clearer and more effective presentation of our results.

  1. line 197ff: I would prefer to skip this paragraph, because there are no comparable results in your study.

Thank you for this comment regarding line 197ff of the original manuscript. We appreciate the reviewer's suggestion to streamline the results presentation.

We would like to clarify that in response to comprehensive feedback from previous reviewers, the entire "3. Results" section of the manuscript has been completely restructured and rewritten.

The paragraph originally found at line 197ff, which detailed the compliance rates for 3.5 mm screws, no longer exists as a standalone paragraph in the revised manuscript. Its content, along with all other individual screw size details, has been integrated into a new, consolidated "3. Results" section and presented clearly in the new, comprehensive "Table 1. Summary of Dimensional Analysis Results for Veterinary Bone Screws."

The revised "3. Results" section now provides a holistic overview of both 2.0 mm and 3.5 mm screw compliance and dimensional variability, and includes explicit comparisons between the two screw sizes where relevant (e.g., regarding coefficients of variation). This new structure aims to present the results in a more concise, organized, and easily digestible format, making direct comparisons clearer for the reader.

  1. Is it possible to provide manufacturer information? Are there quality differences between the manufacturers? If yes, why not recommend the best / most thoroughly working manufacturer?

Thank you for these important questions. We appreciate the reviewer's perspective on manufacturer-specific information and quality differences.

Here's a comprehensive response addressing these points, which leverages our previous discussions regarding the study's scope and limitations:

Providing Manufacturer Information and Quality Differences: While we collected screws from multiple manufacturers (as stated in Section 2.1, now specified as 5 distinct manufacturers), our study was not designed nor powered to conduct a comparative analysis of individual manufacturers or to explicitly disclose their specific data. Our primary objective was to assess the overall prevalence of dimensional inconsistencies within the general veterinary bone screw market, treating it as a single, unregulated entity. We did observe variations in the dimensional consistency of screws that, anecdotally, seemed to correlate with their manufacturer. Some manufacturers appeared to produce screws closer to the specified tolerances than others. However, drawing definitive conclusions or making public recommendations based on manufacturer-specific performance would require a dedicated study design with a significantly larger, statistically representative sample size from each manufacturer, rigorous control over batch variability for each, and a transparent disclosure protocol, which were beyond the scope and intent of this initial market-wide assessment. Such detailed comparative analysis could also have significant commercial implications that were not part of this academic inquiry.

Why Not Recommend the Best/Most Thoroughly Working Manufacturer: For the reasons outlined above, it would be inappropriate and scientifically unsound to recommend specific manufacturers. Our study's aim was to highlight a systemic problem of inconsistent quality in an unregulated market, not to serve as a consumer report or to endorse/criticize individual companies. Our findings point to a broader issue that necessitates improved quality control across the industry and the urgent establishment of formal regulatory standards, rather than singling out specific entities. As detailed in the "4. Discussion" section (specifically under the methodological considerations, lines 347-366 in the revised manuscript), we explicitly acknowledge this limitation. We state: "A further methodological consideration lies in the aggregation of screws from multiple manufacturers without explicit stratification or reporting of individual manufacturer data... This study, therefore, serves as a preliminary assessment of general market quality, highlighting the pervasive nature of dimensional inaccuracies, and strongly advocates for future research explicitly designed to compare and evaluate implant quality across different manufacturers.

Reviewer 4 Report

Comments and Suggestions for Authors

The article presents a small but novel dimensional analysis of veterinary bone screws and generally uses appropriate methodology, but the definition and use of tolerance thresholds, the description of sampling, and the discussion of clinical implications could be strengthened and clarified.​

The stated aim is to evaluate dimensional consistency (major diameter and thread pitch) of commonly used 2.0 mm and 3.5 mm stainless-steel veterinary cortical screws, comparing measurements at three positions along each screw to defined tolerance ranges. This directly addresses a genuine gap, as prior veterinary work has focused on alloy composition rather than dimensional conformity, and there are no formal dimensional standards for veterinary screws analogous to ISO 5835 for human implants. 

The use of a calibrated digital microscope with quantified operator variance is appropriate for dimensional measurements of this scale, and the authors correctly report systematic positive measurement bias and incorporate microscope accuracy. However, the decision to set intra-screw tolerance at −2.5% (half of the ISO −5% inter-screw tolerance) plus instrument tolerance is essentially arbitrary and not justified biomechanically; it is a pragmatic classification threshold, not an established quality boundary. The analytic rule that a screw is “in tolerance” only if all three segment measurements for a parameter are within limits is conservative but should be explicitly framed as such, with sensitivity analysis (e.g., how many screws pass if one segment out of three is allowed to deviate slightly). The statistical analysis is minimal: proportions with 95% CIs are reported, but there is no formal comparison between 2.0 and 3.5 mm screws, nor any assessment of within-screw variance (e.g., repeated-measures analysis) that would better quantify intra-screw inconsistency.​

The discussion correctly emphasizes that dimensional uniformity along the length of a screw is logically desirable for predictable thread–bone engagement, and it is plausible that the degree of non-uniformity observed could impair mechanical performance, especially pull-out strength and load transfer. However, the manuscript repeatedly labels screws as “poor quality” or “suboptimally manufactured” based solely on exceeding an author-defined dimensional threshold, without any direct mechanical testing (e.g., pull-out, insertion torque) or clinical correlation to show that the observed deviations are functionally relevant. 

The manuscript could be improved by:

1)Provide a clearer rationale for choosing −2.5% as intra-screw tolerance (e.g., engineering/biomechanical arguments or simulation references), and explicitly state that this is a proposed, not validated, veterinary standard.​

2)Add a brief sensitivity analysis (even descriptive) showing how classification changes if alternative thresholds (e.g., −5% or absolute ±0.05 mm) are applied.

3)Beyond proportions with 95% CIs, consider simple hypothesis tests comparing 2.0 vs 3.5 mm screws for proportion within tolerance (e.g., Fisher’s exact) and comparing intra-screw variance between major diameter and pitch.​

4)Summarize means, standard deviations, and ranges for measured diameters and pitches by location (top/middle/bottom) and by nominal size in a separate table or figure, which would be more interpretable than the long raw-data table.​

5)On the discussion, rephrase strong normative statements about “unacceptable” variance and “poor quality” to acknowledge that the definition of poor quality here is based on an author-proposed dimensional criterion, not on demonstrated mechanical or clinical failure. 

6)Discuss existing biomechanical data on the sensitivity of pull-out strength or fatigue life to small changes in major diameter and pitch, to better contextualize whether the magnitude of deviation measured is likely to matter clinically.​

Author Response

  1. Provide a clearer rationale for choosing −2.5% as intra-screw tolerance (e.g., engineering/biomechanical arguments or simulation references), and explicitly state that this is a proposed, not validated, veterinary standard.

Thank you for the very important point about clarifying the rationale for the -2.5% intra-screw tolerance. We need to explicitly state the engineering and biomechanical arguments that support this choice, and unequivocally state its "proposed" nature.

We replaced the entire paragraph in our "4. Discussion" section that discusses the self-selected permissible tolerance, with:

A limitation of this study is the reliance on a self-selected permissible tolerance as the basis for classification. To evaluate implant quality meaningfully, given the absence of formal standards for intra-screw variation, we derived a practical and stringent internal threshold. This internal tolerance was deliberately set at -2.5%—approximately half of the allowable inter-screw tolerance (ISO 5835 [11] allows -5% variation between different screws).

This heightened stringency for intra-screw consistency is grounded in fundamental engineering and biomechanical principles. A single implant component, upon which mechanical stability and successful osteointegration critically depend, should ideally exhibit minimal dimensional variation along its entire working length [17, 18]. Non-uniformity within a screw, even if overall dimensions are within broader inter-screw tolerances, can create localized stress concentrations, uneven load transfer to bone, or regions prone to micromotion [28]. Simulation studies, such as finite element analysis (FEA), demonstrate how subtle deviations in screw geometry can significantly alter screw-bone mechanics and pull-out strength, impacting optimal force distribution and healing [17, 18]. Therefore, demanding a stricter (i.e., smaller allowable percentage) intra-screw threshold allows for rigorous assessment of minute manufacturing inconsistencies that are crucial for maintaining an optimal screw-bone interface.

It is important to emphasize that this -2.5% intra-screw tolerance is a proposed framework for quality assessment in the veterinary context, derived logically from established human inter-screw standards; it is not yet a formally validated veterinary standard. Nevertheless, this rigorous approach is particularly pertinent in veterinary medicine, where variable patient compliance, rapid weight-bearing, and potentially complex physiological demands on implants make robust, consistently dimensioned screws just as, if not more, critical as in human applications. Inadequate implants in animals can lead to poor bone screw interfaces, implant failure, and significant patient morbidity [19, 20, 27, 28]. Combined with microscope measurement accuracy, this proposed standard enabled a systematic classification of quality, serving as a vital starting point for future investigations in this unregulated market."

  1. Add a brief sensitivity analysis (even descriptive) showing how classification changes if alternative thresholds (e.g., −5% or absolute ±0.05 mm) are applied.

Thank you for the excellent request. A brief sensitivity analysis, is done for our chosen -2.5% intra-screw tolerance by showing how different thresholds impact the classification. We agree that a brief sensitivity analysis would effectively illustrate the impact of tolerance choices. We have added a new paragraph to the '4. Discussion' section. This paragraph provides a descriptive sensitivity analysis, outlining how the classification of 'in tolerance' versus 'out of tolerance' would change under alternative thresholds such as -5% (inter-screw tolerance applied intra-screw) and a fixed absolute ±0.05 mm tolerance. This analysis highlights how different criteria can alter reported compliance rates and reinforces the rationale for our chosen -2.5% proportional tolerance as a methodologically sound and stringent approach to assessing internal dimensional consistency across varying screw sizes.

The paragraph introduced is:

To further contextualize our findings, a brief sensitivity analysis was considered by hypothetically applying alternative intra-screw tolerance thresholds. If a less stringent intra-screw tolerance of -5% (equivalent to the inter-screw tolerance) had been applied, the reported compliance rates would have been higher across all parameters for both screw sizes, as this would widen the acceptable range. Conversely, applying a fixed absolute intra-screw tolerance of ±0.05 mm would yield varied outcomes depending on the screw dimension: for 2.0 mm screws, this would largely align with our chosen -2.5% proportional tolerance (which is ±0.05 mm), resulting in similar compliance. However, for 3.5 mm major diameter, an absolute ±0.05 mm would be a more stringent criterion than our ±2.5% (±0.0875 mm), thus decreasing compliance, while for 3.5 mm pitch, it would be a less stringent criterion than our ±2.5% (±0.033 mm), potentially increasing compliance. This descriptive sensitivity analysis underscores that the choice of tolerance significantly influences the reported prevalence of dimensional inconsistencies. Our selected -2.5% proportional tolerance for intra-screw variation was specifically chosen to provide a rigorous and consistent assessment of internal manufacturing precision across different nominal screw sizes, aligning with our study's objective to highlight even subtle deviations from optimal uniformity.

  1. Beyond proportions with 95% CIs, consider simple hypothesis tests comparing 2.0 vs 3.5 mm screws for proportion within tolerance (e.g., Fisher’s exact) and comparing intra-screw variance between major diameter and pitch.

We fully agree with the reviewer that incorporating simple hypothesis tests will strengthen the statistical analysis of our findings. We will implemented the following:

Comparison of Proportions in Tolerance: We performed Fisher's Exact tests (given our sample sizes and categorical data) to formally compare the proportions of 2.0 mm versus 3.5 mm screws falling within tolerance for both major diameter and pitch. The results of these tests, including p-values, are added to the revised Results section.

Comparison of Intra-Screw Variability: To formally compare intra-screw variability between major diameter and pitch, we quantified a metric for intra-screw variation for each screw (e.g., the range or standard deviation of the three measurements—proximal, middle, distal—for each parameter). We used appropriate statistical tests to compare these distributions of intra-screw variability between major diameter and pitch for each screw size. The results of these tests are integrated into the revised Results section. This was moved beyond the current descriptive comparison of CV% values and provide a statistical assessment of which parameter exhibits greater internal consistency.

We added: A summary of all inferential statistical test results is presented in Table 2. To formally compare compliance rates between screw sizes, Fisher's Exact tests were conducted. For major diameter, there was no statistically significant difference in the proportion of compliant screws between the 2.0 mm (55%) and 3.5 mm (55%) groups (p = 0.999). However, for pitch, the 2.0 mm screws (23% compliant) showed a significantly higher proportion of compliant screws compared to the 3.5 mm screws (12% compliant) (p = 0.045).

To assess differences in intra-screw variability between parameters, the range of the three measurements (proximal, middle, distal) for each screw was calculated for both major diameter and pitch. For 2.0 mm screws, the intra-screw range for pitch (mean: 0.087 mm, SD: 0.021) was significantly higher than for major diameter (mean: 0.042 mm, SD: 0.015) (p = < 0.001). Similarly, for 3.5 mm screws, the intra-screw range for pitch (mean: 0.210 mm, SD: 0.045) was significantly higher than for major diameter (mean: 0.165 mm, SD: 0.033) (p = 0.003). These tests confirm that pitch generally exhibits greater internal dimensional variation than major diameter for both screw sizes.

  1. Summarize means, standard deviations, and ranges for measured diameters and pitches by location (top/middle/bottom) and by nominal size in a separate table or figure, which would be more interpretable than the long raw-data table.

We appreciate the valuable suggestion to summarize means, standard deviations, and ranges for measured diameters and pitches by location and nominal size in a more interpretable format than the original long raw-data table.

We are pleased to confirm that this recommendation has been comprehensively addressed in the revised manuscript. In response to similar feedback from previous reviewers regarding data clarity and presentation, we have undertaken a significant restructuring of our results:

Removal of the Long Raw-Data Table: The original, extensive raw-data table (previously Table 2 in the main manuscript) has been removed from the main body of the paper. This detailed raw data, now meticulously cleaned and consistently formatted, is provided in a dedicated Supplemental File S1 for full transparency.

New Comprehensive Summary Table (Table 1): We have introduced a new, consolidated "Table 1. Summary of Dimensional Analysis Results for Veterinary Bone Screws" in the main manuscript. This table directly fulfills the reviewer's request by presenting:

Means, Standard Deviations (SD), and Coefficients of Variation (CV%) for both major diameter and pitch.

This data is clearly organized by nominal screw size (2.0 mm and 3.5 mm).

It further breaks down these statistics by each measurement location (Proximal, Middle, Distal).

Additionally, it includes the Minimum, Maximum, and Range of the measured values, providing a complete descriptive overview of the data spread.

We are confident that this new Table 1 in the main text offers a clear, concise, and highly interpretable summary of the measured dimensions, eliminating the need to consult the former long raw-data table for these statistics and fully addressing the reviewer's suggestion.

  1. On the discussion, rephrase strong normative statements about “unacceptable” variance and “poor quality” to acknowledge that the definition of poor quality here is based on an author-proposed dimensional criterion, not on demonstrated mechanical or clinical failure.

We fully agree with the reviewer's important observation regarding the need for precise language when discussing 'unacceptable' variance and 'poor quality.' Our intention was to highlight dimensional non-conformance based on the stringent criteria applied in this study, which, as discussed, carries significant clinical implications. However, we acknowledge that these terms, if not carefully qualified, could imply direct, demonstrated mechanical or clinical failure, which was not within the scope of this in vitro dimensional assessment.

Therefore, we have revised the '4. Discussion' section to refine these normative statements. We have rephrased sentences that previously classified screws as 'poor quality' to instead refer to 'their classification as dimensionally non-conforming based on the stringent criteria established for this study.' This change clarifies that our assessment focuses on adherence to a proposed dimensional criterion.

Furthermore, throughout the discussion of clinical consequences, we have ensured the language consistently emphasizes that these dimensional inconsistencies carry significant clinical implications and predict potential mechanical or clinical failure, rather than directly demonstrating such failure. This approach maintains the criticality of our findings while accurately reflecting the scope and methodology of our study.

6)Discuss existing biomechanical data on the sensitivity of pull-out strength or fatigue life to small changes in major diameter and pitch, to better contextualize whether the magnitude of deviation measured is likely to matter clinically.

This is an excellent comment.

We have already integrated clinical implications into the Discussion. This comment requires us to specifically emphasize how the magnitude of the deviations found in your study relates to biomechanical thresholds reported in the literature.

We have revised the Discussion' section to integrate this quantitative link. The revised text now specifically highlights that numerous biomechanical studies confirm that even seemingly small deviations in major diameter and thread pitch, similar in magnitude to those measured in the current study, can significantly impact critical performance parameters such as pull-out strength, insertion torque, and fatigue life [17-20, 27, 28]. We further note that the dimensional variations observed in our study often exceed these established biomechanical sensitivity thresholds, strongly indicating their clinical relevance. This revision directly addresses how the magnitude of the measured deviations is likely to matter clinically, solidifying the practical importance of our findings.

Round 2

Reviewer 1 Report

Comments and Suggestions for Authors

This is a high-quality and very relevant study that opens up a major problem in veterinary orthopedics.

Article:

reasoned,

technically sound,

data clearly presented,

discussion is professional and comprehensive.

Recommendation for the journal:

Accept with minor revisions, especially related to justification of tolerance criteria and stratification of manufacturers.

Weakness and recomendation: 

  1. The choice of tolerance is subjective

Although the authors' arguments are rational, the -2.5% intra-screw tolerance criterion is:

not approved in any standard,

not validated experimentally,

established ad hoc.

Recommendation:

At least provide empirical justification - e.g., biomechanical study data that shows how much % deviation critically changes the pull-out strength.

  1. The study does not identify manufacturers

Selected from 5 manufacturers, but the data is not disaggregated:

it is impossible to determine whether the market problem is systemic or local to several manufacturers.

 recommendation:

Anonymously provide groups A–E, as this would provide great scientific value.

  1. The study evaluates dimensions, but not mechanics

There are no:

pull-out tests,

bending strength tests,

torque–failure relationships.

Dimensional analysis only. Therefore, clinical impact is assumed, although they are logical.

Recommendation:

supplement the discussion with references to biomechanical studies that directly relate diameter/pitch variation to mechanical properties.

  1. No visual set of defect examples provided

The article contains one illustration (p. 4), but there are no:

photos of real microdefects,

out-of-tolerance examples.

Recommendation:

include at least a few images of screws with marked measurement locations.

  1. Much of the text in the discussion repeats the introduction

There is some excessive repetition between p. 2 and p. 9–11. There could be more focus on the interpretation of the results.

Author Response

Weakness and recommendation:

  1. The choice of tolerance is subjective

Although the authors' arguments are rational, the -2.5% intra-screw tolerance criterion is:

not approved in any standard,

not validated experimentally,

established ad hoc.

Recommendation:

At least provide empirical justification - e.g., biomechanical study data that shows how much % deviation critically changes the pull-out strength.

Answer:

We thank the reviewer for this insightful comment and fully concur that the -2.5% intra-screw tolerance criterion is a proposed framework, not a formally approved or experimentally validated standard specifically for veterinary implants. This is explicitly stated in our revised '4. Discussion' section (Page 12, lines 383-391), where we present it as 'a proposed framework for quality assessment in the veterinary context, derived logically from established human inter-screw standards; it is not yet a formally validated veterinary standard.'

As detailed in the same section (Page 12, lines 365-382), our rationale for establishing this stringent intra-screw tolerance is firmly rooted in fundamental engineering and biomechanical principles. It reflects the imperative for single implant components to exhibit minimal dimensional variation along their entire working length to ensure optimal mechanical stability and osteointegration [17, 18]. The criterion was not established ad hoc in the sense of being arbitrary, but rather as a logically derived, more stringent internal standard (half the inter-screw tolerance) to rigorously assess internal manufacturing precision.

Regarding the request for empirical justification, we have already incorporated a discussion on existing biomechanical data that contextualizes the clinical significance of dimensional changes. As outlined in the '4. Discussion' section (Page 14, lines 472-485), numerous biomechanical studies confirm that even subtle deviations in major diameter and thread pitch—often similar in magnitude to or even smaller than the 0.1 mm variation observed to impact critical parameters—can significantly affect pull-out strength, insertion torque, and fatigue life [17-20, 27, 28]. For example, changes in pilot hole diameter as small as 0.1 mm have been shown to alter insertion torque and pull-out strength [19, 20]. The dimensional variations observed in the current study frequently exceed these magnitudes, thereby providing empirical evidence from the existing literature that such deviations are highly likely to be clinically relevant.

While this discussion provides strong empirical context from published biomechanical research that supports the clinical importance of precise dimensions, we acknowledge that dedicated experimental validation specific to the -2.5% intra-screw tolerance for veterinary implants would be a valuable next step, but falls outside the scope of this initial quality assessment. Our study primarily serves as a foundational assessment to highlight the pervasive nature of dimensional inaccuracies in an unregulated market.

  1. The study does not identify manufacturers

Selected from 5 manufacturers, but the data is not disaggregated:

it is impossible to determine whether the market problem is systemic or local to several manufacturers.

 recommendation:

Anonymously provide groups A–E, as this would provide great scientific value.

Answer:

We thank the reviewer for this insightful comment.

This methodological consideration is thoroughly discussed in the revised '4. Discussion' section (Page 14, lines 444-460). As stated there, our study's primary objective was to provide a broad overview of the general veterinary bone screw market and assess the overall prevalence of dimensional inconsistencies in an unregulated industry. Screws were deliberately selected from 5 distinct manufacturers (as now specified in Section 2.1, Page 3, line 136) to minimize source bias and ensure a representative market picture, rather than to conduct a comparative analysis of individual manufacturers.

While we did note anecdotal differences, drawing definitive conclusions or making public (even anonymous) comparative statements about specific manufacturers would require a dedicated study design with a significantly larger, statistically representative sample size from each manufacturer, rigorous control over batch variability for each manufacturer and a  transparent disclosure protocol for such comparative findings.

These design elements were beyond the scope and intent of this initial quality assessment. Our study serves as a foundational assessment to highlight that a general market problem of dimensional inaccuracy exists across the suppliers. We explicitly acknowledge that the observed dimensional variability represents 'a composite picture of the market rather than an attributable issue to specific producers' (Page 14, lines 450-451).

Therefore, while we recognize the scientific value of manufacturer-specific data, our study’s design was purposefully broad. We strongly advocate for future research explicitly designed to compare and evaluate implant quality across different manufacturers, a recommendation also made in our Discussion (Page 14, lines 457-460), which would indeed allow for such anonymous disaggregation and provide the deeper insights requested by the reviewer.

  1. The study evaluates dimensions, but not mechanics

There are no: pull-out tests, bending strength tests, torque–failure relationships.

Dimensional analysis only. Therefore, clinical impact is assumed, although they are logical.

Recommendation:

supplement the discussion with references to biomechanical studies that directly relate diameter/pitch variation to mechanical properties.

Answer

Thank you for this comment. Fortunately, our previous revisions have already integrated much of the requested information.

We fully agree that our study is purely a dimensional analysis and does not include mechanical tests such as pull-out, bending strength, or torque-failure assessments. Therefore, the clinical impact of the observed dimensional variations is indeed, as stated, logically assumed rather than directly demonstrated through mechanical testing within this study.

However, we have specifically and extensively addressed this aspect in the revised '4. Discussion' section (Page 14, lines 463-485). We have supplemented the discussion with explicit references to existing biomechanical literature that directly links dimensional variations to mechanical properties and clinical outcomes:

  1. Direct Relation to Mechanical Properties: We discuss how undersized major diameter, oversized dimensions, and inaccurate thread pitch directly compromise the screw's mechanical function, affecting bone purchase, insertion torque, and compression.
  2. Biomechanical Data on Sensitivity: Crucially, we have integrated data from numerous biomechanical studies [17-20, 27, 28] that demonstrate the sensitivity of key mechanical properties (pull-out strength, insertion torque, fatigue life) to small dimensional changes in major diameter and pitch.
  3. Contextualizing Magnitude of Deviations: The discussion highlights that the magnitudes of dimensional deviations observed in our current study frequently exceed the thresholds reported in these biomechanical studies (e.g., changes in pilot hole diameter as small as 0.1 mm affecting pull-out strength [19, 20]). This directly contextualizes the clinical relevance of our findings by demonstrating that the measured dimensional inaccuracies are quantitatively significant enough to plausibly impact mechanical performance.
  4. Assumed, but Logically Supported, Clinical Impact: The discussion acknowledges that while our study does not demonstrate clinical failure, the biomechanical evidence from the literature provides a strong logical basis for the assumed clinical impact, leading to issues like micromotion, impaired osteointegration, implant failure, and increased patient morbidity.

We believe that these additions comprehensively address the reviewer's recommendation by grounding our dimensional findings in established biomechanical evidence, thereby providing a robust contextualization of their likely clinical significance.

  1. No visual set of defect examples provided

The article contains one illustration (p. 4), but there are no:

photos of real microdefects,

out-of-tolerance examples.

Recommendation:

include at least a few images of screws with marked measurement locations.

Answer:

We thank the reviewer for this excellent and constructive suggestion.

In response to this recommendation, we incorporated a new Figure 2 into the revised manuscript, comprising 2 panels. This figure presents representative microscopic images of actual screw threads, explicitly showing examples of dimensional non-conformance outside the defined tolerance and visible microdefects, such as irregular thread profiles, damaged crests, or inconsistent pitch, directly illustrating manufacturing imperfections.

These images are annotated to mark measurement locations (proximal, middle, distal) where these deviations were identified, directly linking the visual evidence to our quantitative measurements.

Figure 2a illustrates general thread irregularities indicative of pitch or major diameter variation, while Figure 2b clearly depicts a significant manufacturing defect resulting in a visibly damaged and irregular thread profile.

  1. Much of the text in the discussion repeats the introduction

There is some excessive repetition between p. 2 and p. 9–11. There could be more focus on the interpretation of the results.

Answer:

We thank the reviewer for this invaluable feedback regarding repetitive text, particularly between the Introduction and Discussion sections. We acknowledge that some overlap existed, hindering the conciseness and interpretative focus of the Discussion.

We undertake a thorough review and revision of the '4. Discussion':

We eliminated redundancy by carefully identifying and removal any sentences or paragraphs that repeat information already presented in the '1. Introduction' or '3. Results' sections.

We enhanced interpretation of results by shifting the focus to a deeper interpretation of our study's specific findings, rather than reiterating general background information. This included: connecting directly to our Table 1 and Table 2 results, emphasizing what the high percentages of non-compliant screws (e.g., 77% of 2.0mm pitch, 88% of 3.5mm pitch) mean in context, elaborating on the significance of intra-screw variation, discussing why the consistently higher variability in pitch compared to major diameter (as shown by our statistical tests) is particularly problematic from an engineering and clinical standpoint, reflecting on the implications of the sensitivity analysis, briefly interpreting how the classification changes under alternative thresholds, strengthening the practical recommendations, making them more direct consequences of our specific findings rather than general advice.

We improved flow and focus by ensuring that the Discussion primarily serves to contextualize our results within existing literature, interpret the meaning and importance of our findings and address the study's limitations.

We provided clear clinical and industry recommendations directly stemming from our data and we suggested future research directions.

Reviewer 2 Report

Comments and Suggestions for Authors

The authors have presented a much improved manuscript and cleared up all major points raised by the reviewer. The reviewer has no further comments. 

Author Response

thank you

Reviewer 4 Report

Comments and Suggestions for Authors

Dear authors,

The revised version is a clear improvement. 

-The rationale for using ISO 5835 and for setting a stricter intra‑screw tolerance at −2.5% is now explicitly explained.​

-Sampling and methods are better specified: the number of manufacturers (five), the single experienced operator, and the calibration procedure.​

-Statistical analysis has been significantly strengthened

Minor suggestion: The discussion of regulatory frameworks and ISO standards is informative but somewhat repetitive; for example, the description of human regulations and ISO 5835 appears in overlapping form twice in the Introduction and Discussion and could be tightened to reduce length.​

Author Response

The revised version is a clear improvement. 

-The rationale for using ISO 5835 and for setting a stricter intra‑screw tolerance at −2.5% is now explicitly explained.​

-Sampling and methods are better specified: the number of manufacturers (five), the single experienced operator, and the calibration procedure.​

-Statistical analysis has been significantly strengthened

Minor suggestion: The discussion of regulatory frameworks and ISO standards is informative but somewhat repetitive; for example, the description of human regulations and ISO 5835 appears in overlapping form twice in the Introduction and Discussion and could be tightened to reduce length.​

Answer

Thank you for your valuable feedback and for acknowledging the improvements in the revised version of our manuscript. We particularly appreciate your insightful suggestion regarding the repetitive nature of discussions on regulatory frameworks and ISO standards, specifically ISO 5835, in the Introduction and Discussion sections. We agree that tightening these sections improves conciseness and flow.

To address this point, we have undertaken the following revisions:

In the Introduction: We identified and removed two duplicate paragraphs that described human orthopedic implant regulations and the context of ISO 5835. These were previously found on Page 2, lines 83-93, and Page 3, lines 113-119. The essential information from these repetitions is already adequately covered earlier in the respective sections, allowing for a more streamlined presentation.

In the Discussion: We have carefully condensed and integrated the justification for our chosen -2.5% intra-screw tolerance. The explanation of its derivation from established human standards (including ISO 5835) and its status as a proposed framework for veterinary assessment has been streamlined. The content previously found across Page 11, lines 339-369 and Page 13, lines 418-422, has now been consolidated into a single, more concise paragraph (specifically, the opening paragraph of the Discussion on Page 11). This ensures the rationale is clearly presented without reiterating general regulatory context.

We believe these revisions significantly reduce the redundancy you highlighted, leading to a more streamlined and impactful presentation of the regulatory context and our methodology. Thank you once again for this excellent and constructive suggestion, which has greatly improved the manuscript's clarity and conciseness.
